# Adeno-Associated Viral Vectors as a Tool for Large Gene Delivery to the Retina

**DOI:** 10.3390/genes10040287

**Published:** 2019-04-09

**Authors:** Ivana Trapani

**Affiliations:** 1Telethon Institute of Genetics and Medicine (TIGEM), 80078 Pozzuoli, Italy; trapani@tigem.it; Tel.: +39-081-1923-0684; 2Medical Genetics, Department of Translational Medicine, Federico II University, 80131 Naples, Italy

**Keywords:** AAV, retina, gene therapy, dual AAV

## Abstract

Gene therapy using adeno-associated viral (AAV) vectors currently represents the most promising approach for the treatment of many inherited retinal diseases (IRDs), given AAV’s ability to efficiently deliver therapeutic genes to both photoreceptors and retinal pigment epithelium, and their excellent safety and efficacy profiles in humans. However, one of the main obstacles to widespread AAV application is their limited packaging capacity, which precludes their use from the treatment of IRDs which are caused by mutations in genes whose coding sequence exceeds 5 kb. Therefore, in recent years, considerable effort has been made to identify strategies to increase the transfer capacity of AAV vectors. This review will discuss these new developed strategies, highlighting the advancements as well as the limitations that the field has still to overcome to finally expand the applicability of AAV vectors to IRDs due to mutations in large genes.

## 1. Introduction

The eye is an ideal target for gene therapy thanks to its small and enclosed structure, relative immune privilege and easy accessibility [1,2]. This has boosted attempts at developing gene therapy approaches for the treatment of a large number of inherited retinal diseases (IRDs) over the recent decades [3,4]. Confirmation of the advancements in the retinal gene therapy field came in the last two years with the approval of the first gene therapy product for an IRD, Luxturna [5]—an adeno-associated viral (AAV) vector-based therapy for a form of Leber Congenital Amaurosis [6]—in the US, first, and then in Europe. The recombinant AAV vector on which Luxturna is based is the most widely used vector for retinal gene delivery. AAV are small (25 nm), nonenveloped, icosahedral viruses belonging to the Parvoviridae family [7]. They package a linear single-stranded DNA genome of ~4.7 kb, flanked by two 145 bp long palindromic inverted terminal repeats (ITRs) [7]. These ITRs form hairpin-loop secondary structures at the strand termini and are the only viral sequences that are retained in cis in the recombinant AAV vector genome [7]. Recombinant vectors based on AAV have fast become popular in the gene therapy field because of their excellent safety profile and low immunogenicity which allows for long-term expression of the therapeutic gene, at least in post-mitotic tissues, so that most experimental therapy studies require only a single vector administration. Additionally, dozens of different AAV variants have been identified thus far, each of them with unique transduction characteristics. This allows the user to select the most appropriate AAV serotype to transduce the retinal cell layer of interest. Indeed, following subretinal delivery, virtually all the AAV serotypes tested efficiently transduced the retinal pigment epithelium (RPE), while the levels of transduction of photoreceptors, which are the main therapeutic target cells in most IRDs, varied significantly among different serotypes [4,8]. AAV5, AAV7, AAV8 and AAV9 serotypes have all been demonstrated to efficiently transduce photoreceptors [4,8]. Additional serotypes with increased retinal transduction abilities have also been identified through either rational design or directed evolution [4,8]. This is one of the most attractive features of AAV vectors for retinal gene therapy, since alternative, both non-viral and viral, vectors tested thus far have shown more limited transduction abilities of adult photoreceptors [2,9]. For all the above described reasons, AAV have been used in many successful preclinical and clinical studies [3,4]. Clinical trial data collected over a decade have confirmed the overwhelming safety of AAV vectors delivered intraocularly and shown many instances of efficacy in treating previously incurable IRDs.

However, one of the main limitations to a broader application of AAV vectors for retinal gene therapy is their packaging capacity, which is restricted to approximately 5 kb of DNA [10]. This vector capacity is a critical issue, given the fact that approximately 6% of all human proteins have a coding sequence (CDS) that exceeds 4 kb [11] and that, in addition to the CDS of the therapeutic gene and the ITRs, a gene therapy vector needs to include, as a minimum, a promoter and a polyadenylation signal (polyA). Thus, the treatment of disorders caused by mutations in genes over 4 kb in size, including those causative of common IRDs, is currently not achievable using standard AAV vector-mediated approaches. The development of strategies to overcome AAV packaging limitation has therefore become a key area of research within the gene therapy field.

## 2. Strategies for Large Gene Delivery

Two types of strategies have been developed for large gene delivery via AAV: one is based on the “forced” packaging of oversized genomes (i.e., larger than 5 kb) in a single AAV vector (oversized AAV vectors); the other relies on the delivery of portions of large transgenes in two AAV vectors, which recombine through various mechanisms in the target cell, leading to the reconstitution of the full-length gene (dual AAV vectors) (Table 1).

### 2.1. Oversized AAV Vectors

Several research groups have tried to encapsidate large genes in a single AAV vector [12,13,14]. These “oversized” AAV vectors have been found to successfully express full-length proteins in vitro and in the retina of IRD models to levels which led to significant and stable improvement of the phenotype [12,15]. However, the genome contained in oversized AAV vectors was found to be not a pure population of intact large-sized genomes but rather a mixture of genomes highly heterogeneous in size [14,16,17,18,19]. Thus, it was proposed that full-length protein expression from oversized AAV vectors was achieved, following infection, through the re-assembly of truncated genomes in the target cell nucleus [14,16,17,18,19]. The efficiency of the transduction of oversized AAV vectors in the retina in comparison to alternative platforms for large gene delivery (i.e., dual AAV vectors, discussed below) has been assessed in various studies and found to be variable. Whereas some studies found considerably high levels of transgene expression from oversized AAV vectors [14,15], others showed efficient large protein reconstitution only upon dual AAV vector delivery [20,21]. Both the design and purification process of oversized AAV vectors were hypothesized to be critical for the success of the strategy, as the use of transgenes slightly above 5 kb can give rise to genomes with longer overlaps compared to the use of transgenes largely exceeding AAV cargo capacity, and this can drive more efficient re-assembly of oversized AAV vectors. Along this line, it was shown that the fractionation of oversized AAV vector preparations can be explored to promote selection of the genomes with the highest transduction properties in the final viral preparation [14]. However, despite the optimization and ability of this strategy to reconstitute large genes expression in vivo, consistently shown in various studies, the heterogenous nature of oversized AAV genomes poses major safety concerns, limiting their further application in clinical settings.

### 2.2. Dual AAV Vectors

An alternative strategy for AAV-mediated large gene delivery is the generation of dual AAV vectors. In this strategy, large transgenes are split into two separate AAV vectors that, upon co-infection of the same cell, reconstitute the expression of a full-length gene via intermolecular recombination between the two AAV vector genomes. This ideally doubles AAV cargo capacity, allowing delivery of transgenes up to about 9 kb. Various dual AAV vector strategies have been developed (referred to as trans-splicing [22], overlapping [23] and hybrid [24] dual AAV vector strategies), which differ in the mechanism they use to reconstitute the transgene.

#### 2.2.1. Trans-Splicing Dual AAV Vectors

The trans-splicing approach relies on the natural ability of AAV ITRs to concatemerize in order to reconstitute full-length genomes [22,25]. In this approach, the two vectors carry two separate halves of the transgene, without regions of sequence overlap; the 5’-half vector has a splice donor (SD) signal at the 3’ end of the AAV genome, while the 3’-half vector carries a splice acceptor (SA) signal at the 5’ end of the AAV genome (Figure 1).

This allows splicing of the concatemerized ITR structure that forms in the middle of the therapeutic CDS following tail-to-head concatemerization of the two AAV genomes to obtain a single large mRNA molecule. This approach was first tested about 20 years ago, and historically represents the first developed approach for AAV-mediated large gene delivery. Since then, many studies have shown the efficacy of this strategy to reconstitute large genes. The major limitation of this platform, however, is that concatemerization can occur between any of the ITR of the two vectors. This may lead to the formation of both forms of circular monomers of each AAV, as well as two-vector linear concatemers in a number of orientations of which only one (i.e., tail-to-head concatemer) is productive to restore full-length gene expression [26]. Attempts at favoring the formation of concatemers in the correct orientation have been made (as discussed in the “Limitations of dual AAV vectors” paragraph). An additional limiting step of trans-splicing vectors is splicing across the ITR junction, the efficiency of which is dependent on both selection of the optimal exon–exon junction for splitting the large therapeutic gene [27] as well as the efficiency of splicing across the ITR structure [28]. To overcome the first issue, synthetic SD and SA signals have been developed, which mediate high rates of splicing independently of the gene that needs to be delivered [29]. Yet, since the sequence surrounding the splicing signals has an impact on splicing efficiency, careful selection of the splitting point is required.

#### 2.2.2. Overlapping Dual AAV Vectors

In the overlapping approach, the transgene is split into two halves sharing homologous overlapping sequences, such that the reconstitution of the large gene expression cassette relies on homologous recombination [23] (Figure 2).

As it has been designed, the overlapping approach is the simplest in design and requires less foreign or artificial DNA elements when compared to the other approaches. However, as the success of this strategy is critically dependent upon the ability of the overlapping region to mediate efficient homologous recombination, much work is needed to determine the optimal CDS overlapping region to be used for each transgene. Furthermore, data obtained so far have also highlighted that the success of this strategy is dependent on the retinal cell type being targeted, since the efficiency of the repair mechanism on which overlapping dual AAV vectors rely for large gene reconstitution is tissue dependent, as discussed below.

#### 2.2.3. Hybrid Dual AAV Vectors

To overcome the main limitations of the previously described platforms (i.e., the lack of preference for directional tail-to-head concatemerization of the trans-splicing approach and the need for optimization of the CDS overlap for each transgene in the overlapping approach), a third transgene-independent dual AAV approach was developed: the hybrid dual AAV vectors. This approach is a combination of the trans-splicing and overlapping approaches, as it is based on the addition of a highly recombinogenic exogenous sequence to the trans-splicing vectors in order to increase recombination efficiency [24]. This recombinogenic sequence is placed downstream of the SD signal in the 5’-half vector and upstream of the SA signal in the 3’-half vector, so to be spliced out from the mRNA after recombination and transcription (Figure 3).

The hybrid dual AAV approach is potentially more effective than the other dual AAV vector approaches, since full-length gene reconstitution can occur through both homologous recombination mediated by the highly recombinogenic exogenous sequence as well as concatemerization through the ITRs [24]. The recombinogenic sequences used thus far to induce the recombination between hybrid dual AAV vectors have been derived from regions of either the alkaline phosphatase gene (AP) [24,30] or the F1 phage genome (AK) [21]. The inclusion of the exogenous sequence allows the promotion of high levels of homologous recombination between the two vector genomes, independently of the transgene to be delivered. However, similarly to the trans-splicing approach, the sequences surrounding the splicing signals still have an impact on splicing efficiency. Thus, careful selection of the splitting point is recommended to achieve maximal efficacy of large gene reconstitution.

## 3. The Choice of the Best Platform for Large Gene Delivery to the Retina

The efficacy of both oversized and dual AAV vectors in the retina has been evaluated in a number of studies using different reporter and therapeutic genes, such as *ABCA4* and *MYO7A* mutated in Stargardt disease (STGD1) [31] and Usher syndrome type 1B (USH1B) [32], respectively. However, literature describing these platforms is often conflicting. Initial studies in the retina reported a better performance of oversized AAV vectors compared to dual AAV strategies [14,15]. These results, however, might be due to both design and purification processes, which favor the generation of oversized vectors with high transduction properties [14], as well as to the less than optimal design of the dual AAV platform that was used as a comparison. One study, as an example, relied on the use of overlapping dual AAV vectors with a large region of overlap (1365 bases) that had not been optimized and, therefore, might potentially have a low efficiency of recombination [15]. Reconstitution from overlapping dual AAV vectors has also been found to occur at variable levels in different studies [15,20,21,26,33]. The most critical aspect of an overlapping dual AAV vector strategy is the event of recombination between the two halves of the transgene. This is influenced by both the sequence of the transgene and the cell type that is targeted, since different cell types could possibly deploy different DNA repair mechanisms. Some studies have found that long regions of overlap may lead to higher levels of transgene reconstitution [26]. However, it has recently been shown that optimization of the overlapping region is a prerequisite to achieve sustained levels of transgene expression in photoreceptors, since the efficiency of reconstitution is not directly proportional to the length of the regions of overlap [33]. It has been suggested that if the regions are too short, they might not be able to efficiently mediate interactions with the opposing viral genome, whereas longer regions of overlap may be less available for such interactions due to secondary structure formation. In line with this hypothesis, a screening of overlapping regions ranging from 23 to 1173 bp identified an overlap of 207–505 bp as the best performing for overlapping dual AAV-mediated reconstitution of *ABCA4* at therapeutic levels [33]. Thus, optimization of the overlapping region is essential to achieve sustained levels of transgene expression in photoreceptors. The targeted tissue also plays an important role in the success of the overlapping dual AAV approach since homologous recombination is typically associated with dividing cells, while low levels of homologous recombination are found in post-mitotic cells as neurons [34]. Along this line, studies have reported inefficient transduction of photoreceptors mediated by overlapping dual AAV vectors [15,21,26], whilst more efficient reconstitution was found in the RPE [21]. Other groups, however, have found efficient transduction of photoreceptors using overlapping dual AAV vectors [20,33], highlighting that the identification of highly recombinogenic regions of overlap in the transgene overcomes the limitations related to the inability of specific cell types to mediate efficient homologous recombination [33].

More consensus on the efficacy of trans-splicing and hybrid dual AAV vectors can be found in literature. A number of studies have indeed shown the ability of these strategies to reconstitute large transgenes in the retina [20,21,26,35,36] at levels which were higher compared to the other dual AAV strategies tested side by side [20,21,26], and which resulted in improvement of the retinal phenotype of animal models of IRDs [21,37]. This is possibly due to a more limited requirement of the optimization of these platforms compared to the others, since joining of the two halves of the transgene, with a discrete nature, occurs through the ITRs and/or a region of overlap known to be highly recombinogenic. Notably, the success obtained in the delivery of the large *MYO7A* gene to the retina [21] has led to the planning of a Phase I/II clinical trial, which will test the safety and efficacy of the hybrid dual AAV platform developed in the retina of USH1B patients (https://cordis.europa.eu/project/rcn/212674_it.html). Importantly, the results of this trial will definitively shed light on the efficiency of dual AAV vectors-mediated large gene delivery in the human retina.

Prompted by the success shown by dual AAV strategies, researchers have attempted at further expanding AAV cargo capacity in the retina up to 14 kb by adding a third vector to the dual system, generating triple AAV vectors [38]. This was found to be achievable, but at the expense of efficiency. Indeed, the levels of transduction achieved in the retina of a mouse model of Alstrom syndrome with triple AAV vectors have led to only a modest and transient improvement of the phenotype [38]. On the other hand, the levels of transduction mediated by triple AAV vectors in the large pig retina were found to be significantly higher than in the mouse retina, as also observed with dual AAV vectors [38]. These results bode well for further optimization of this platform.

## 4. Limitations of Dual AAV Vectors

Currently, all the dual AAV vector approaches have shown similar issues: variable success and expression of unwanted truncated products from single half-vectors. For all dual AAV platforms to be successful, a cell must necessarily be co-infected by at least one AAV vector including the 5’- and one including the 3’-half of the expression cassette. We and others have shown that co-transduction by two AAV vectors is quite efficient in the small subretinal space [11,21,36], which thus represents a favorable environment for developing dual AAV vector-based gene therapy approaches.

So far, however, all the studies performed have shown that none of the dual AAV approach matches the levels of expression achieved with a single AAV vector [21,26,36]. Various strategies have been explored to increase the efficiency of dual AAV vector-mediated large gene reconstitution.

One option is to increase vector dose and/or use AAV serotypes with higher tropism for the target cells in order to maximize rates of co-infection by both half vectors. A recent study has, however, suggested that an increase in vector dose does not proportionally correlate with increased levels of protein expression in the retina [26]. This suggests that, once efficient co-transduction is achieved, a further increase in vector genome amounts does not provide significant advantages [26]. Attempts at achieving higher levels of transduction by using alternative AAV serotypes have not been found consistently to result in higher transduction levels. Some studies have shown that use of capsid-engineered AAV variants with higher retinal transduction abilities, as tyrosine mutants capsids [39], led to higher levels of transgene expression from overlapping dual AAV vectors compared to naturally occurring AAV serotypes [20,33]. However, delivery of hybrid dual AAV vectors using an in-silico designed, synthetic vector (Anc80L65), which has also been shown to transduce retinal cells with a higher efficiency than AAV8 [40], led to almost identical levels of protein reconstitution compared to dual AAV8 vectors [26].

Another approach explored to increase transduction levels from dual AAV vectors has been maximizing the chances of both trans-splicing and hybrid AAV vectors to generate concatemers in the productive orientation, by forcing concatemerization of the ITRs, through the use of vectors carrying heterologous ITRs (i.e., ITR from different AAV serotypes at the opposite ends of the viral genome) [41]. Indeed, by generating trans-splicing vectors with heterologous ITR from serotypes 2 and 5 it has been shown that it is possible to reduce both the ability of each vector to form circular monomers and to increase directional tail-to-head concatemerization. This resulted in increased levels of transgene reconstitution compared to the use of vectors with homologous ITRs [41,42]. However, we have later shown that inclusion of heterologous ITRs in hybrid dual AAV vectors does not provide a significant advantage in full-length transgene reconstitution over the use of vectors with homologous ITRs [37]. This is consistent with the idea that hybrid dual AAV concatemerization is already partially driven in the correct orientation by the presence of highly recombinogenic regions. An additional strategy which has been used to direct AAV vectors concatemerization in the proper orientation is the use of a single-strand DNA oligonucleotide displaying homology to both of the distinct AAV genomes [43]. Alternatively, strategies that can improve dual AAV vector transduction efficiency by positively modulating AAV transduction steps, as the delivery of kinase inhibitors along with AAV vectors, have also been tested [44].

Another major drawback of dual AAV vectors, observed in some studies, is the production of truncated protein products from each of the single AAV vectors [20,21,33,37]. We and others have shown that, both in vitro and in the retina, truncated proteins from the 5′ half vector that contains the promoter sequence and/or from the 3′ half vector, due to the low promoter activity of the ITR, are produced. This issue can however be efficiently overcome by the use of the CL1 degron, a C-terminal destabilizing peptide that shares structural similarities with misfolded proteins and is thus recognized by the ubiquitination system [45,46]. Inclusion of this short (16 amino acids in length) degron mediates selective degradation of the truncated product from the 5′ half vector [37], without either affecting full-length protein reconstitution or significantly reducing the packaging capacity of the platform. More recently, McClements et al. have shown how the design of dual AAV vectors can also influence production of truncated proteins by the generation of unintended cryptic translation start sites and/or polyA signals [33]. Thus, the design of these platforms requires multiple considerations and adaptation, which may include codon optimizations to remove cryptic genetic signals. Furthermore, given the expression of such unwanted protein products, confirmation of the safety of dual AAV vectors is an important open question. While our preliminary data have shown no evident alterations of retinal morphology and functionality in mouse and pig eyes injected with dual AAV vectors [21,37], formal toxicity studies are required to elucidate this aspect.

## 5. Alternative Strategies to Allow AAV-Mediated Large Gene Delivery

Additional strategies to deliver large transgenes via AAV vectors are being actively investigated. Attempts at identifying AAV vectors with expanded cargo capacity, based on either protein libraries and directed evolution [47] or site directed mutagenesis to add positively-charged residues at lumenally exposed sites within the capsid [48], have been described. Alternatively, it has been shown that oversized AAV2 vector genomes can be effectively packaged in the capsid of human Bocavirus 1 (HBoV1) [49,50], an autonomous parvovirus relative of AAV, with a 5.5 kb genome. Testing of these vectors in the retina might lead to the identification of novel suitable vectors for large gene delivery.

The development of different short regulatory elements has also been attempted to reduce the size of the expression cassette and allow delivery of transgenes that exceed the AAV packaging capacity [51,52,53,54,55]. However, this often led to reduced levels of transgene expression. The combination of short synthetic enhancers and promoters was found to be useful for providing increased levels of expression of large transgenes [56]. Other studies have however shown that, despite optimization, some transgenes were more difficult than others to reconstitute from oversized AAV vectors when using short promoters [57].

The use of cDNA encoding for truncated versions of large proteins, which retain their functionality (i.e., a minigene), has also been achieved with some success [58]. However, all these approaches still cannot be easily applied to a large number of genes that exceed the AAV cargo capacity, since extensive optimization and testing would be required for each one of them.

## 6. Conclusions and Outlook

The growing number of clinical trials that show good safety and efficacy of the subretinal delivery of AAV vectors are contributing to the establishment of AAV as vectors of choice for retinal gene transfer. Expanding AAV cargo capacity over 5 kb is however a prerequisite to allow this platform to be used as a tool for the efficient delivery of a larger number of therapeutic genes. Recent proof-of-concept studies that used dual and triple AAV vectors to deliver large genes to the retina have shown that it is feasible to transfer genes with a CDS larger than 5 kb. Yet, these studies have highlighted that there is no one-fits-all dual AAV vector system, since dual AAV approaches have shown different relative efficiency in different studies. Clearly, the tissue being targeted, as well as the transgene that needs to be delivered, drastically influences transduction efficiency. Thus, careful design of the platform for each therapeutic application is required to achieve maximal efficacy. The planned clinical trial for USH1B will help defining whether the levels of expression achieved with dual AAV vectors are therapeutically relevant in humans. While the need of manufacturing two or more vectors to treat each disorder might represent a challenge of dual/triple AAV platforms, yet the retina is a favorable tissue for development of these approaches due to the fact that it requires delivery of only a small amount of vector. This reduces the total amount of vectors that needs to be produced.

Retinal transduction with multiple AAV vectors has been shown to reach lower levels compared to a single AAV vector. These levels were not sufficient to result in therapeutic efficacy for some diseases [38]. Consequently, alternative strategies should be explored.

Systems that rely on mechanisms different than those exploited by dual AAV vectors for large gene reconstitution might be investigated, including trans-splicing of pre-mRNAs [59] or intein-mediated protein trans-splicing [60]. Genome editing is also a rapidly expanding field of research, and could represent an interesting option for correction of mutations in genes whose delivery through AAV vectors is precluded by the large CDS size. A number of aspects for this approach however still need to be further explored. First, in the retina, where homologous recombination occurs at low rates, genome editing tools for the precise correction of a mutation will most probably need to exploit alternative repair mechanisms such as non-homologous end joining used for homology-independent targeted integration [61]. The efficiency of such approaches in the retina is still unknown. Secondly, the delivery of genome editing tools in post-mitotic tissues, such as the retina, might not be as safe as delivery in more proliferative tissues, considering the fact that their expression will persist long term after a single subretinal injection.

In conclusion, important steps forward have been made towards the treatment of IRDs due to mutations in large genes, which now seems an achievable goal. The optimization of these and the newly emerging platforms will allow expansion of the number of IRDs that are treatable using AAV-mediated gene therapy.

## Figures and Tables

**Figure 1 genes-10-00287-f001:**
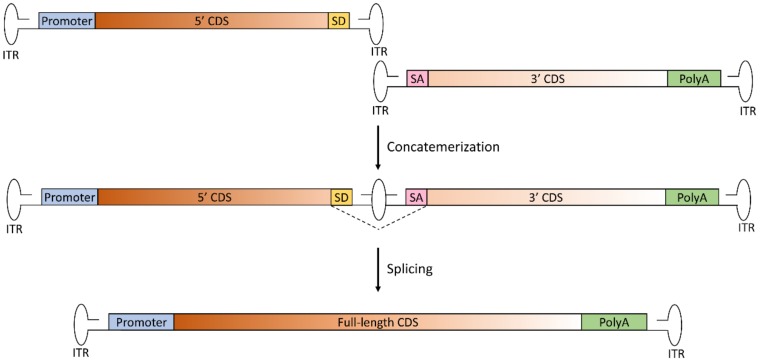
Schematic representation of the trans-splicing dual AAV approach for large gene reconstitution. The first vector includes the promoter, the 5’-half of the coding sequence (CDS) and the splicing donor (SD) signal; the second vector includes the splicing acceptor (SA) signal, the 3’-half of the CDS and the polyadenylation signal (PolyA). Concatemerization of the two vectors, involving the right-hand inverted terminal repeat (ITR) of the first vector and the left-hand ITR of the second vector, reconstitutes the full-length gene. After transcription, splicing leads to the removal of the ITR structure at the junction point, with restoration of the full-length, mature RNA of the transgene.

**Figure 2 genes-10-00287-f002:**
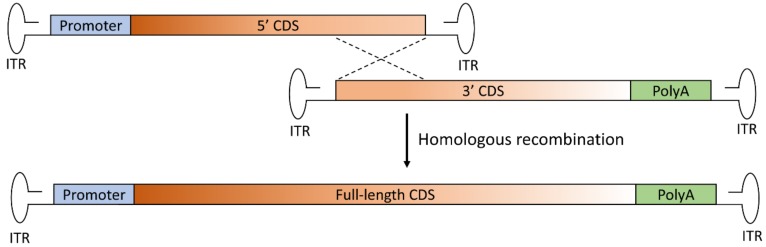
Schematic representation of the overlapping dual AAV approach for large gene reconstitution. The first vector includes the promoter and the 5’-half of the coding sequence (CDS) and the second vector includes the 3’-half of the CDS and the polyadenylation signal (PolyA). A portion of the sequence of the large transgene is repeated in both vectors (at the 3’ end of the CDS of the first vector and at the 5’ end of the CDS of the second vector). Thus, the full-length transgene expression cassette is reconstituted through homologous recombination of the overlapping regions in the two vectors. ITR: inverted terminal repeat.

**Figure 3 genes-10-00287-f003:**
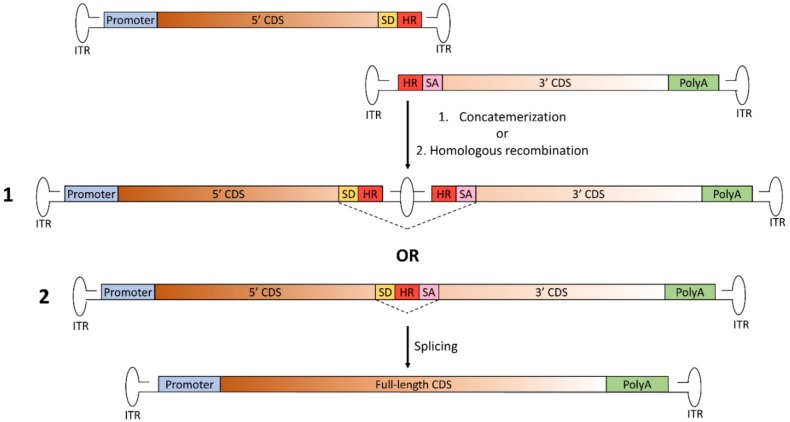
Schematic representation of the hybrid dual AAV approach for large gene reconstitution. The first vector includes the promoter, the 5’-half of the coding sequence (CDS), the splicing donor (SD) signal and the highly recombinogenic exogenous sequence (HR); the second vector includes the highly recombinogenic exogenous sequence, the splicing acceptor (SA) signal, the 3’-half of the CDS and the polyadenylation signal (PolyA). Joining of the two AAV vector genomes to reconstitute the full-length gene can occur through either: 1. concatemerization of the two vectors through the inverted terminal repeats (ITR), as for trans-splicing dual AAV vectors; or 2. homologous recombination mediated by the region of homology included in both vectors. In both cases, after transcription, splicing leads to the removal of the junction point, with restoration of the full-length, mature RNA of the transgene.

**Table 1 genes-10-00287-t001:** Adeno-associated viral (AAV) vector-based strategies for large gene delivery.

Strategy	Advantages	Limitations
Oversized AAV	No need to identify optimal splitting points/region of overlap	Genome highly heterogeneous in size
Trans-Splicing Dual AAV	Genomes with discrete nature	Non-directional concatemerization (with only one concatemer being productive) Need to identify optimal splitting points Efficiency dependent on splicing across the inverted terminal repeat (ITR) junction Potential production of shorter protein products
Overlapping Dual AAV	Genomes with discrete nature No additional foreign or artificial DNA elements required	Need to identify the optimal region of overlap for efficient homologous recombination Potential production of shorter protein products
Hybrid Dual AAV	Genomes with discrete nature Relies on two mechanisms for transgene reconstitution Transgene-independent efficacy of recombination	Need to identify optimal splitting points Efficiency dependent on splicing across the ITR junction Potential production of shorter protein products

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
