# Peer review of "Adeno-Associated Viral Vectors as a Tool for Large Gene Delivery to the Retina"

_genes, 2019, doi:10.3390/genes10040287_

Reviewer 1 Report

The review manuscript “Adeno-Associated Viral Vectors as a Tool for Large Gene Delivery to the Retina” discusses currently available approaches to package and deliver genes larger than 5 kb using AAV. This review gives an overview on using AAV as a gene delivery vector for inherited retinal diseases by discussing the safety and utility of AAV vectors as well as their limitations. This manuscript would be strengthened by more specific details about how these vector engineering strategies impact delivery to the retina.

 Major comments:

 1)    There are multiple review articles about dual vector strategies and there are multiple review articles about AAV delivery to the retina. This review needs to better address an unexplored niche.

 2)    Figures 1, 2, and 3 show progressive advancements in the same idea. It would be helpful to know how much better the efficiency for generating a full-length CDS each strategy achieves. Are there examples of each strategy in the retina?

 3)    There is repetitive text. For example, the paragraph beginning on line 128 and the paragraph beginning on line 154 have regions of identical phrasing. Section 2.2 could be shortened. Figure 1 strategy could be described in detail and Figure 2 and 3 strategies should focus on the differences to strategy 1. If those differences are relevant to the retina, then that difference should be stressed.

 4)    Different serotypes of AAV have slightly different packaging capacities (ie. Reference #12 Grieger et al JVI 2005). A brief discussion of serotypes with eye tropism and their relative packaging capacity would be of interest. Currently, there are no AAV capsids discussed in this manuscript.

 5)    The author should speculate on other strategies that may lead to packaging larger transgenes. For example, there is an idea of increasing packaging capacity through directed evolution or site directed mutagenesis1. Bocavirus or other Parvoviruses with naturally larger packaging capacities may be made into vectors and have retina tropism.

1[Abstract] Tiffany M, Kay MA. Expanded Packaging Capacity of AAV by Lumenal Charge Alteration. Molecular Therapy. Vol 24. S99-S100

 6)    Efforts to shrink the size of the expression cassette could be have its own section in the review. Internal deletions to cDNA and short polyAs are mentioned, but efforts to generate short synthetic promoters for cell specific expression could also be discussed.

 7)    Beginning with line 259, please give an example of a degradation signal that can be included to prevent truncated proteins from expressing. Additionally, please comment on the size of this signal and how this may affect the packaging capacity.

Author Response

I thank the reviewer for his/her suggestions to improve the manuscript, which were taken into account as described below.

1)      There are multiple review articles about dual vector strategies and there are multiple review articles about AAV delivery to the retina. This review needs to better address an unexplored niche.

I agree with the reviewer that there are various reviews about dual AAV vectors and gene delivery to the retina. However, while paragraph 2, being descriptive of the available strategies for large gene delivery, might be partially redundant with other articles, no recent reviews specifically focused on application of these platforms in the retina are available, to my knowledge. Both in paragraphs 3 and 4, describing efficacy and limitations of these platforms, I have focused mostly on data obtained in the retina. Given the variable efficacy of these platforms in different tissue, I think this review might be helpful to researchers approaching to large gene delivery specifically to the retina, in order to identify the key aspects to be taken into account when designing platforms to transduce this tissue. According to the suggestion of the reviewer, I have tried to better underline the retinal-related nature of the findings described.

2)      Figures 1, 2, and 3 show progressive advancements in the same idea. It would be helpful to know how much better the efficiency for generating a full-length CDS each strategy achieves. Are there examples of each strategy in the retina?

As mentioned above, paragraph 3 includes only examples of the efficacy of each strategy in the retina. Based on the reviewer suggestion, I have now underlined more in detail the results from studies in the retina which performed side-by-side comparison of the strategies, to help the reader understanding their relative efficiency (page 7, lines 233-236). Yet, as also underlined in the manuscript, it is difficult to provide a definitive conclusion on which is the most efficient platform to reconstitute large genes in the retina, since the efficacy of these strategy was found to be variable across different studies, possibly because of differences in platforms design.

3)      There is repetitive text. For example, the paragraph beginning on line 128 and the paragraph beginning on line 154 have regions of identical phrasing. Section 2.2 could be shortened. Figure 1 strategy could be described in detail and Figure 2 and 3 strategies should focus on the differences to strategy 1. If those differences are relevant to the retina, then that difference should be stressed.

I apologize for the misunderstanding. Text in page 3 lines 99-104, page 4 lines 128-133 and page 5 lines 154-162 (of the previous version of the manuscript), were the legends to Figures 1, 2 and 3 respectively. I have now more clearly reformatted and separated them from the main text, which now includes only relevant details on the differences between each strategy, as also suggested by the reviewer.

4)    Different serotypes of AAV have slightly different packaging capacities (ie. Reference #12 Grieger et al JVI 2005). A brief discussion of serotypes with eye tropism and their relative packaging capacity would be of interest. Currently, there are no AAV capsids discussed in this manuscript.

I have added a paragraph on AAV serotypes (page 1, lines 38-44 and page 2, lines 49-50), as suggested, as well as discussion of studies in which use of different AAV serotypes has impacted on transduction efficiency of the discussed platforms (page 8, lines 268-275). However, since there is not much consensus in literature regarding the impact of AAV serotype on packaging of large genes [Grieger et al, J Virol. 2005 Aug;79(15):9933-44; Wang et al, Hum Gene Ther Methods. 2012 Aug;23(4):225-33; Hirsch et al, Mol Ther. 2010 Jan;18(1):6-8; Allocca et al, J Clin Invest. 2008 May;118(5):1955-64; Wu et al, Mol Ther. 2010 Jan;18(1):80-6], and there is not a definitive conclusion, I have not mentioned this aspect in the manuscript. 

5)    The author should speculate on other strategies that may lead to packaging larger transgenes. For example, there is an idea of increasing packaging capacity through directed evolution or site directed mutagenesis1. Bocavirus or other Parvoviruses with naturally larger packaging capacities may be made into vectors and have retina tropism. 

1[Abstract] Tiffany M, Kay MA. Expanded Packaging Capacity of AAV by Lumenal Charge Alteration. Molecular Therapy. Vol 24. S99-S100

As suggested, I have added discussion of these interesting approaches aimed at increasing AAV packaging capacity, as well as of attempts at producing chimeric vectors based on AAV and Bocaviruses (page 9, lines 316-322).

6)    Efforts to shrink the size of the expression cassette could be have its own section in the review. Internal deletions to cDNA and short polyAs are mentioned, but efforts to generate short synthetic promoters for cell specific expression could also be discussed.

A new section on “Alternative strategies to allow AAV-mediated large gene delivery” has been added to the manuscript. This includes a paragraph on strategies to reduce the size of the expression cassette, which has been modified as suggested (page 9, lines 323-329).

7)    Beginning with line 259, please give an example of a degradation signal that can be included to prevent truncated proteins from expressing. Additionally, please comment on the size of this signal and how this may affect the packaging capacity.

Text has been modified according to the comment (page 8, lines 298-303).

Reviewer 2 Report

This review consists of description of rAAV vector use to deliver large transgenes for correction of genetic disorders and follows-up multiple other journal reviews generated recently on dual rAAV vector strategies. Within this manuscript, the author focuses on rAAV-mediated transgene delivery to eye and in particular to photoreceptors and retinal pigment epithelium where mutations are a cause for many inherited disorders causing blindness. Though able to provide safe and efficient gene delivery, the major issues of this vector system is its limited DNA packaging capacity and this has been a challenge for treatment two of the retinal disorders in particular, Stargardt disease and Usher syndrome which both have mutations in large transgenes.

 Overall, the review is well written and the diagrams describing the dual vector strategies are clear and helpful. The text  summarizes the majority of the efforts in the field including the authors’ own work.  Most of the dual rAAV vector strategies have been covered by many recent reviews but this manuscript contains some updates in the rapidly evolving field. These include potential solutions on how to design vectors to minimize production of truncated proteins and possible ways to enhance vector concatemerization required for trans-splicing and hybrid dual vector strategy. The author concludes that there is no one-fits-all dual AAV vector system which is consistent with others in the field. Furthermore, as the author states, the translatability of current animal work with dual rAAV vectors will be tested soon in the clinical trial for Ushers syndrome.

 Some of the following aspect could be included in the manuscript to as food for thought:

1.     Does efficacy of the dual rAAV vector strategies require potentially high vector doses to be delivered? The dual vector strategy, as the name implies, requires at least one copy of each vector to generate a full-length transgene. Though the eye is considered an immune privileged site immune response issues could arise with high vector doses due to unintended vector leakage. With strategies requiring concatemerization, the need for multiple vector genomes is likely even higher as majority of rAAV vector genomes tend to be converted into monomeric forms.

 2.     The practicality of manufacturing two or even three vectors for a single disorder? Generation of a therapy that requires multiple rAAV vectors will be a challenging product concept with each individual vector requiring its own quality control and assays. It is likely that the cost of rAAV manufacturing will be reduced over time with improved manufacturing and hopefully the medical need will continue to drive development of more efficient dual vectors to justify need to multiple vectors.

Author Response

I thank the reviewer for his/her positive comments. I have taken into account both of his/her suggestions in order to further improve the manuscript, as follows:

 1.       Does efficacy of the dual rAAV vector strategies require potentially high vector doses to be delivered? The dual vector strategy, as the name implies, requires at least one copy of each vector to generate a full-length transgene. Though the eye is considered an immune privileged site immune response issues could arise with high vector doses due to unintended vector leakage. With strategies requiring concatemerization, the need for multiple vector genomes is likely even higher as majority of rAAV vector genomes tend to be converted into monomeric forms.

As we and others have shown, the small volume of the subretinal space favors infection and transduction of the same cell by two independent AAV vectors. Thus, most of the proof-of-concept studies thus far performed used doses in the range of those used in proof-of-concept studies which relied on single AAV vectors to deliver therapeutic transgenes [108-109 genome copies/eye; Trapani et al, EMBO Mol Med. 2014 Feb;6(2):194-211; Hirsch et al., Mol Ther. 2013 Dec;21(12):2205-16; Carvalho et al, Front Neurosci. 2017 Sep 8;11:503]. These dual AAV doses resulted in no evident toxicity in our preliminary studies in mice and pigs, as discussed in the manuscript (page 8, line 308 and page 9, lines 311-312). Additionally, a recent study has suggested that an increase in vector dose does not proportionally correlate with increased levels of protein expression in the retina, and use of vectors with high tropism for photoreceptors and RPE cells might be sufficient to achieve efficient co-transduction at intermediate vector dose ranges. I have now discussed all these aspects in the manuscript (page 8, lines 264-268).
2.       The practicality of manufacturing two or even three vectors for a single disorder? Generation of a therapy that requires multiple rAAV vectors will be a challenging product concept with each individual vector requiring its own quality control and assays. It is likely that the cost of rAAV manufacturing will be reduced over time with improved manufacturing and hopefully the medical need will continue to drive development of more efficient dual vectors to justify need to multiple vectors.

I agree with the point raised by the reviewer about the issue of manufacturing two separate vectors to treat each disease. Yet, I think this might be not a major limitation to development of dual AAV vectors for treatment of inherited retinal diseases, given that only a small amount of vector is delivered to the subretinal space, thus one vector preparation can be theoretically used to treat a large number of patients. I have now included a discussion of this important aspect in the manuscript (page 9, lines 347-351).

Reviewer 3 Report

Well-written and focused manuscript. Provides the necessary information about the current strategies aiming to overcome one of the main obstacles of AAV vectors as a gene therapy tool. Packaging capacity of AAV remains as a major hurdle for using AAV as a therapeutic strategy for several diseases. The unique features of eye anatomy facilitate the application of a dual-AAV approach in which the author has long and considerable expertise with many publications in this area. 

Major comments:

1 - I would recommend the author to move the description of the strategies to the figure legend of the corresponding figure (i.e. lines 99 to 104, page 3 to Figure Legend 1). This will facilitate the understanding of the described strategy by checking the figure. 

2 - The role of the promoter in oversized vectors has been described previously by several authors, especially in AAV for neurodegenerative diseases.  A small paragraph pointing out this effect would enrich the manuscript overall. 

Minor comments:

The acronym AAV generally refers to an adeno-associated virus, meanwhile, when referring to the genetic tool I will recommend to keep it homogeneous through the manuscript as "AAV vector". 

Line 2 mention unique features of the eye but does not explain which are these features in detail.

A summary table containing the different strategies, benefits, and weaknesses will enrich considerably the manuscript and provide a quick glance for the readers.

  The hybrid dual AAV vector deserved a separate section on its own that emphasize the and novelty of this strategy.  

Tripe AAV vector strategy should be move from section 3 to section 2. As it refers to another already a validated and reported strategy.

Line 133. ITR: inverted terminal repeats should be removed and added to the corresponding figure legends.

Author Response

I thank the reviewer for appreciating the manuscript as well as for his/her suggestions to improve it. Accordingly, the manuscript has been modified, as described below.

 Major comments:

 1 - I would recommend the author to move the description of the strategies to the figure legend of the corresponding figure (i.e. lines 99 to 104, page 3 to Figure Legend 1). This will facilitate the understanding of the described strategy by checking the figure.

I apologize for the misunderstanding. Text in page 3, lines 99-104, page 4, lines 128-133, and page 5, lines 154-162 (of the previous version of the manuscript) were indeed the legends to Figures 1, 2 and 3 respectively. I have now more clearly reformatted and separated them from the main text.

 2 - The role of the promoter in oversized vectors has been described previously by several authors,especially in AAV for neurodegenerative diseases.  A small paragraph pointing out this effect would enrich the manuscript overall.

A new section on “Alternative strategies to allow AAV-mediated large gene delivery” has been added to the manuscript. This includes a paragraph on strategies to reduce the size of the expression cassette, in which also the impact of promoter selection in oversized vectors has been discussed, as suggested (see page 9, lines 323-329).

 Minor comments:

The acronym AAV generally refers to an adeno-associated virus, meanwhile, when referring to the genetic tool I will recommend to keep it homogeneous through the manuscript as "AAV vector".

I have modified the text according to the suggestion.

Line 2 mention unique features of the eye but does not explain which are these features in detail.

I have clarified the most important features of the eye which make it an ideal target tissue for retinal gene therapy.

A summary table containing the different strategies, benefits, and weaknesses will enrich considerably the manuscript and provide a quick glance for the readers.

I have included a summary table describing advantages and limitations of each platform in order to help the reader to perform comparison among them (Table 1 in page 2).

  The hybrid dual AAV vector deserved a separate section on its own that emphasize the and novelty of this strategy. 

I agree with the reviewer that the hybrid dual AAV vector was an innovative strategy, as it was designed to overcome the main limitations of both of the previously described platforms and develop a transgene-independent approach, as underlined in the manuscript (page 5, lines 164-167). To better highlight the differences between the various strategies, separate sections for each of the dual AAV vector approaches have now been made.

Tripe AAV vector strategy should be move from section 3 to section 2. As it refers to another already a validated and reported strategy.

I agree with the reviewer that triple AAV vectors are more considered an expansion of the dual AAV vector platforms. However, since either the trans-splicing, or the overlapping, or the hybrid mechanisms, as well as a combination of two of them, has been explored to reconstitute full-length genes [Koo et al, Hum Gene Ther. 2014 Feb;25(2):98-108; Lostal et al, Hum Gene Ther. 2014 Jun;25(6):552-62; Maddalena et al, Mol Ther. 2018 Feb 7;26(2):524-541], it might be not completely exhaustive to include this strategy in one of the paragraphs of section 2. Additionally, since this review is focused on application of platforms for large gene delivery to the retina, where only one study using triple AAV vectors exists, I think it is more appropriate to cite this platform in paragraph 3, which is aimed at providing a general overview of the available strategies and their efficiency.

Line 133. ITR: inverted terminal repeats should be removed and added to the corresponding figure legends.

As mentioned in a previous comment, text in page 4 lines 128-133 (of the previous version of the manuscript) was indeed meant to be part of the Figure legend. I have now properly formatted the legends in order to clarify this.

Round  2

Reviewer 1 Report

The author has addressed my prior concerns.